# Scalable End-to-End Autonomous Vehicle Testing via Rare-event Simulation

**Matthew O'Kelly**[*]
University of Pennsylvania
mokelly@seas.upenn.edu

**Aman Sinha**[*]
Stanford University
amans@stanford.edu

**Hongseok Namkoong**[*]
Stanford University
hnamk@stanford.edu

**John Duchi**
Stanford University
jduchi@stanford.edu

**Russ Tedrake**
Massachusetts Institute of Technology
russt@mit.edu

## Abstract

While recent developments in autonomous vehicle (AV) technology highlight substantial progress, we lack tools for rigorous and scalable testing. Real-world testing, the *de facto* evaluation environment, places the public in danger, and, due to the rare nature of accidents, will require billions of miles in order to statistically validate performance claims. We implement a simulation framework that can test an entire modern autonomous driving system, including, in particular, systems that employ deep-learning perception and control algorithms. Using adaptive importance-sampling methods to accelerate rare-event probability evaluation, we estimate the probability of an accident under a base distribution governing standard traffic behavior. We demonstrate our framework on a highway scenario, accelerating system evaluation by 2-20 times over naive Monte Carlo sampling methods and 10-300P times (where P is the number of processors) over real-world testing.

## 1 Introduction

Recent breakthroughs in deep learning have accelerated the development of autonomous vehicles (AVs); many research prototypes now operate on real roads alongside human drivers. While advances in computer-vision techniques have made human-level performance possible on narrow perception tasks such as object recognition, several fatal accidents involving AVs underscore the importance of testing whether the perception and control pipeline—when considered as a *whole system*—can safely interact with humans. Unfortunately, testing AVs in real environments, the most straightforward validation framework for system-level input-output behavior, requires prohibitive amounts of time due to the rare nature of serious accidents [49]. Concretely, a recent study [29] argues that AVs need to drive "hundreds of millions of miles and, under some scenarios, hundreds of billions of miles to create enough data to clearly demonstrate their safety." Alteratively, formally verifying an AV algorithm's "correctness" [34, 2, 47, 37] is difficult since all driving policies are subject to crashes caused by other drivers [49]. It is unreasonable to ask that the policy be safe under *all* scenarios. Unfortunately, ruling out scenarios where the AV should not be blamed is a task subject to logical inconsistency, combinatorial growth in specification complexity, and subjective assignment of fault.

Motivated by the challenges underlying real-world testing and formal verification, we consider a probabilistic paradigm—which we call a *risk-based framework*—where the goal is to evaluate the *probability of an accident* under a base distribution representing standard traffic behavior. By assigning learned probability values to environmental states and agent behaviors, our risk-based

---

[*]Equal contribution

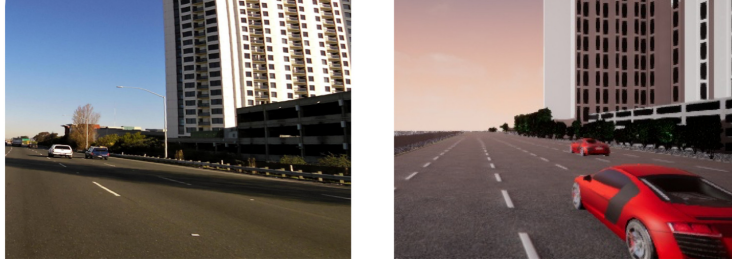

**Figure 1.** Multi-lane highway driving on I-80: (left) real image, (right) rendered image from simulator

framework considers performance of the AV's policy under a data-driven model of the world. To efficiently evaluate the probability of an accident, we implement a photo-realistic and physics-based simulator that provides the AV with perceptual inputs (*e.g.* video and range data) and traffic conditions (*e.g.* other cars and pedestrians). The simulator allows parallelized, faster-than-real-time evaluations in varying environments (*e.g.* weather, geographic locations, and aggressiveness of other cars).

Formally, we let $P_0$ denote the base distribution that models standard traffic behavior and $X \sim P_0$ be a realization of the simulation (*e.g.* weather conditions and driving policies of other agents). For an objective function $f : \mathcal{X} \to \mathbb{R}$ that measures "safety"—so that low values of $f(x)$ correspond to dangerous scenarios—our goal is to evaluate the probability of a dangerous event

$$p_\gamma := \mathbb{P}_0(f(X) \leq \gamma) \tag{1}$$

for some threshold $\gamma$. Our risk-based framework is agnostic to the complexity of the ego-policy and views it as a black-box module. Such an approach allows, in particular, deep-learning based perception systems that make formal verification methods intractable.

An essential component of this approach is to estimate the base distribution $P_0$ from data; we use public traffic data collected by the US Department of Transportation [36]. While such datasets do not offer insights into how AVs interact with human agents—this is precisely why we design our simulator—they illustrate the range of standard human driving behavior that the base distribution $P_0$ must model. We use imitation learning [45, 41, 42, 22, 6] to learn a generative model for the behavior (policy) of environment vehicles; unlike traditional imitation learning, we train an ensemble of models to characterize a distribution of human-like driving policies.

As serious accidents are rare ($p_\gamma$ is small), we view this as a *rare-event simulation* [4] problem; naive Monte Carlo sampling methods require prohibitively many simulation rollouts to generate dangerous scenarios and estimate $p_\gamma$. To accelerate safety evaluation, we use adaptive importance-sampling methods to learn alternative distributions $P_\theta$ that generate accidents more frequently. Specifically, we use the cross-entropy algorithm [44] to iteratively approximate the optimal importance sampling distribution. In contrast to simple classical settings [44, 55] which allow analytic updates to $P_\theta$, our high-dimensional search space requires solving convex optimization problems in each iteration (Section 2). To address numerical instabilities of importance sampling estimators in high dimensions, we carefully design search spaces and perform computations in logarithmic scale. Our implementation produces 2-20 times as many rare events as naive Monte Carlo methods, independent of the complexity of the ego-policy.

In addition to accelerating evaluation of $p_\gamma$, learning a distribution $P_\theta$ that *frequently* generates realistic dangerous scenarios $X_i \sim P_\theta$ is useful for engineering purposes. The importance-sampling distribution $P_\theta$ not only efficiently samples dangerous scenarios, but also ranks them according to their likelihoods under the base distribution $P_0$. This capability enables a deeper understanding of failure modes and prioritizes their importance to improving the ego-policy.

As a system, our simulator allows fully distributed rollouts, making our approach orders of magnitude cheaper, faster, and safer than real-world testing. Using the asynchronous messaging library ZeroMQ [21], our implementation is fully-distributed among available CPUs and GPUs; our rollouts are up to 30P times faster than real time, where P is the number of processors. Combined with the cross-entropy method's speedup, we achieve 10-300P speedup over real-world testing.

In what follows, we describe components of our open-source toolchain, a photo-realistic simulator equipped with our data-driven risk-based framework and cross-entropy search techniques. The

toolchain can test an AV as a *whole system*, simulating the driving policy of the ego-vehicle by viewing it as a black-box model. The use of adaptive-importance sampling methods motivates a unique simulator architecture (Section 3) which allows real-time updates of the policies of environment vehicles. In Section 4, we test our toolchain by considering an end-to-end deep-learning-based ego-policy [9] in a multi-agent highway scenario. Figure 1 shows one configuration of this scenario in the real world along with rendered images from the simulator, which uses Unreal Engine 4 [17]. Our experiments show that we accelerate the assessment of rare-event probabilities with respect to naive Monte Carlo methods as well as real-world testing. We believe our open-source framework is a step towards a rigorous yet scalable platform for evaluating AV systems, with the broader goal of understanding how to reliably deploy deep-learning systems in safety-critical applications.

## 2 Rare-event simulation

To motivate our risk-based framework, we first argue that formally verifying correctness of a AV system is infeasible due to the challenge of defining "correctness." Consider a scenario where an AV commits a traffic violation to avoid collision with an out-of-control truck approaching from behind. If the ego-vehicle decides to avoid collision by running through a red light with no further ramifications, is it "correct" to do so? The "correctness" of the policy depends on the extent to which the traffic violation endangers nearby humans and whether any element of the "correctness" specification explicitly forbids such actions. That is, "correctness" as a binary output is a concept defined by its exceptions, many elements of which are subject to individual valuations [10].

Instead of trying to verify correctness, we begin with a continuous measure of safety $f : \mathcal{X} \to \mathbb{R}$, where $\mathcal{X}$ is space of traffic conditions and behaviors of other vehicles. The prototypical example in this paper is the minimum time-to-collision (TTC) (see Appendix A for its definition) to other environmental agents over a simulation rollout. Rather than requiring safety for all $x \in \mathcal{X}$, we relax the deterministic verification problem into a probabilistic one where we are concerned with the probability under standard traffic conditions that $f(X)$ goes below a safety threshold. Given a distribution $P_0$ on $\mathcal{X}$, our goal is to estimate the rare event probability $p_\gamma := P_0(f(X) \leq \gamma)$ based on simulated rollouts $f(X_1), \ldots, f(X_n)$. As accidents are rare and $p_\gamma$ is near 0, we treat this as a rare-event simulation problem; see [11, 4, Chapter VI] for an overview of this topic.

First, we briefly illustrate the well-known difficulty of naive Monte Carlo simulation when $p_\gamma$ is small. From a sample $X_i \overset{\text{iid}}{\sim} P_0$, the naive Monte Carlo estimate is $\widehat{p}_{N,\gamma} := \frac{1}{N} \sum_{i=1}^{N} \mathbf{1}\{f(X_i) \leq \gamma\}$. As $p_\gamma$ is small, we use relative accuracy to measure our performance, and the central limit theorem implies the relative accuracy is approximately

$$\left| \frac{\widehat{p}_{N,\gamma}}{p_\gamma} - 1 \right| \overset{\text{dist}}{\approx} \sqrt{\frac{1 - p_\gamma}{N p_\gamma}} |Z| + o(1/\sqrt{N}) \text{ for } Z \sim \mathcal{N}(0, 1).$$

For small $p_\gamma$, we require a sample of size $N \gtrsim 1/(p_\gamma \epsilon^2)$ to achieve $\epsilon$-relative accuracy, and if $f(X)$ is light-tailed, the sample size must grow exponentially in $\gamma$.

**Cross-entropy method** As an alternative to a naive Monte Carlo estimator, we consider (adaptive) importance sampling [4], and we use a model-based optimization procedure to find a good importance-sampling distribution. The optimal importance-sampling distribution for estimating $p_\gamma$ has the conditional density $p^\star(x) = \mathbf{1}\{f(x) \leq \gamma\} p_0(x)/p_\gamma$, where $p_0$ is the density function of $P_0$: as $p_0(x)/p^\star(x) = p_\gamma$ for all $x$ satisfying $\mathbf{1}\{f(x) \leq \gamma\}$, the estimate $\widehat{p}^\star_{N,\gamma} := \frac{1}{N} \sum_{i=1}^{N} \frac{p_0(X_i)}{p^\star(X_i)} \mathbf{1}\{f(X_i) \leq \gamma\}$ is exact. This sampling scheme is, unfortunately, *de facto* impossible, because we do not know $p_\gamma$. Instead, we use a parameterized importance sampler $P_\theta$ and employ an iterative model-based search method to modify $\theta$ so that $P_\theta$ approximates $P^\star$.

The cross-entropy method [44] iteratively tries to find $\theta^\star \in \operatorname{argmin}_{\theta \in \Theta} D_{\text{kl}}(P^\star \| P_\theta)$, the Kullback-Leibler projection of $P^\star$ onto the class of parameterized distributions $\mathcal{P} = \{P_\theta\}_{\theta \in \Theta}$. Over iterations $k$, we maintain a surrogate distribution $q_k(x) \propto \mathbf{1}\{f(x) \leq \gamma_k\} p_0(x)$ where $\gamma_k \geq \gamma$ is a (potentially random) proxy for the rare-event threshold $\gamma$, and we use samples from $P_\theta$ to update $\theta$ as an approximate projection of $Q$ onto $\mathcal{P}$. The motivation underlying this approach is to update $\theta$ so that $P_\theta$ upweights regions of $\mathcal{X}$ with low objective value (*i.e.* unsafe) $f(x)$. We fix a quantile level $\rho \in (0, 1)$—usually we choose $\rho \in [0.01, 0.2]$—and use the $\rho$-quantile of $f(X)$ where $X \sim P_{\theta_k}$

---

**Algorithm 1** Cross-Entropy Method

---

1: Input: Quantile $\rho \in (0,1)$, Stepsizes $\{\alpha_k\}_{k\in\mathbb{N}}$, Sample sizes $\{N_k\}_{k\in\mathbb{N}}$, Number of iterations $K$
2: Initialize: $\theta_0 \in \Theta$
3: **for** $k = 0,1,2,\ldots,K-1$ **do**
4:      Sample $X_{k,1},\ldots,X_{k,N_k} \overset{\text{iid}}{\sim} P_{\theta_k}$
5:      Set $\gamma_k$ as the minimum of $\gamma$ and the $\rho$-quantile of $f(X_{k,1}),\ldots,f(X_{k,N_k})$
6:      $\theta_{k+1} = \operatorname{argmax}_{\theta\in\Theta}\left\{\alpha_k\theta^\top D_{k+1} + (1-\alpha_k)\theta^\top\nabla A(\theta_k) - A(\theta)\right\}$

---

as $\gamma_k$, our proxy for the rare event threshold $\gamma$ (see [23] for alternatives). We have the additional challenge that the $\rho$-quantile of $f(X)$ is unknown, so we approximate it using i.i.d. samples $X_i \sim P_{\theta_k}$. Compared to applications of the cross-entropy method [44, 55] that focus on low-dimensional problems permitting analytic updates to $\theta$, our high-dimensional search space requires solving convex optimization problems in each iteration. To address numerical challenges in computing likelihood ratios in high-dimensions, our implementation carefully constrains the search space and we compute likelihoods in logarithmic scale.

We now rigorously describe the algorithmic details. First, we use natural exponential families as our class of importance samplers $\mathcal{P}$.

**Definition 1.** *The family of density functions $\{p_\theta\}_{\theta\in\Theta}$, defined with respect to base measure $\mu$, is a* natural exponential family *if there exists a sufficient statistic $\Gamma$ such that $p_\theta(x) = \exp(\theta^\top\Gamma(x) - A(\theta))$ where $A(\theta) = \log\int_{\mathcal{X}}\exp(\theta^\top\Gamma(x))d\mu(x)$ is the log partition function and $\Theta := \{\theta \mid A(\theta) < \infty\}$.*

Given this family, we consider idealized updates to the parameter vector $\theta_k$ at iteration $k$, where we compute projections of a mixture of $Q_k$ and $P_{\theta_k}$ onto $\mathcal{P}$

$$
\begin{aligned}
\theta_{k+1} &= \underset{\theta\in\Theta}{\operatorname{argmin}}\, D_{\text{kl}}\left(\alpha_k Q_k + (1-\alpha_k)P_{\theta_k}\|P_\theta\right)\\
&= \underset{\theta\in\Theta}{\operatorname{argmax}}\left\{\alpha_k\mathbb{E}_{Q_k}[\log p_\theta(X)] + (1-\alpha_k)\mathbb{E}_{\theta_k}[\log p_\theta(X)]\right\}\\
&= \underset{\theta\in\Theta}{\operatorname{argmax}}\left\{\alpha_k\theta^\top\mathbb{E}_{Q_k}[\Gamma(X)] + (1-\alpha_k)\theta^\top\nabla A(\theta_k) - A(\theta)\right\}. \quad (2)
\end{aligned}
$$

The term $\mathbb{E}_{Q_k}[\Gamma(X)]$ is unknown in practice, so we use a sampled estimate. For $X_{k,1},\ldots,X_{k,N_k}\overset{\text{iid}}{\sim} P_{\theta_k}$, let $\gamma_k$ be the $\rho$-quantile of $f(X_{k,1}),\ldots,f(X_{k,N_k})$ and define

$$
D_{k+1} := \frac{1}{N_k}\sum_{i=1}^{N_k}\frac{q_k(X_{k,i})}{p_{\theta_k}(X_{k,i})}\Gamma(X_{k,i}) = \frac{1}{N_k}\sum_{i=1}^{N_k}\frac{p_0(X_{k,i})}{p_{\theta_k}(X_{k,i})}\mathbf{1}\left\{f(X_{k,i})\leq\gamma_k\right\}\Gamma(X_{k,i}). \quad (3)
$$

Using the estimate $D_{k+1}$ in place of $\mathbb{E}_{Q_k}[\Gamma(X)]$ in the idealized update (2), we obtain Algorithm 1. To select the final importance sampling distribution from Algorithm 1, we choose $\theta_k$ with the lowest $\rho$-quantile of $f(X_{k,i})$. We observe that this choice consistently improves performance over taking the last iterate or Polyak averaging. Letting $\theta_{\text{ce}}$ denote the parameters for the importance sampling distribution learned by the cross-entropy method, we sample $X_i \overset{\text{iid}}{\sim} P_{\theta_{\text{ce}}}$ and use $\widehat{p}_{N,\gamma} := \frac{1}{N}\sum_{i=1}^N\frac{p_0(X_i)}{p_{\theta_{\text{ce}}}(X_i)}\mathbf{1}\left\{f(X_i)\leq\gamma\right\}$ as our final importance-sampling estimator for $p_\gamma$.

In the context of our rare-event simulator, we use a combination of Beta and Normal distributions for $P_\theta$. The sufficient statistics $\Gamma$ include (i) the parameters of the generative model of behaviors that our imitation-learning schemes produce and (ii) the initial poses and velocities of other vehicles, pedestrians, and obstacles in the simulation. Given a current parameter $\theta$ and realization from the model distribution $P_\theta$, our simulator then (i) sets the parameters of the generative model for vehicle policies and draws policies from this model, and (ii) chooses random poses and velocities for the simulation. Our simulator is one of the largest-scale applications of cross-entropy methods.

## 3 Simulation framework

Two key considerations in our risk-based framework influence design choices for our simulation toolchain: (1) learning the base distribution $P_0$ of nominal traffic behavior via data-driven modeling, and (2) testing the AV as a *whole system*. We now describe how our toolchain achieves these goals.

## 3.1 Data-driven generative modeling

While our risk-based framework (cf. Section 2) is a concise, unambiguous measure of system safety, the rare-event probability $p_\gamma$ is only meaningful insofar as the base distribution $P_0$ of road conditions and the behaviors of other (human) drivers is estimable. Thus, to implement our risk-based framework, we first learn a base distribution $P_0$ of nominal traffic behavior. Using the highway traffic dataset NGSim [36], we train policies of human drivers via imitation learning [45, 41, 42, 22, 6]. Our data consists of videos of highway traffic [36], and our goal is to create models that imitate human driving behavior even in scenarios distinct from those in the data. We employ an ensemble of generative adversarial imitation learning (GAIL) [22] models to learn $P_0$. Our approach is motivated by the observation that reducing an imitation-learning problem to supervised learning—where we simply use expert data to predict actions given vehicle states—suffers from poor performance in regions of the state space not encountered in data [41, 42]. Reinforcement-learning techniques have been observed to improve generalization performance, as the imitation agent is able to explore regions of the state space in simulation during training that do not necessarily occur in the expert data traces.

Generically, GAIL is a minimax game between two functions: a discriminator $D_\phi$ and a generator $G_\xi$ (with parameters $\phi$ and $\xi$ respectively). The discriminator takes in a state-action pair $(s, u)$ and outputs the probability that the pair came from real data, $\mathbb{P}(\text{real data})$. The generator takes in a state $s$ and outputs a conditional distribution $G_\xi(s) := \mathbb{P}(u \mid s)$ of the action $u$ to take given state $s$. In our context, $G_\xi(\cdot)$ is then the (learned) policy of a human driver given environmental inputs $s$. Training the generator weights $\xi$ occurs in a reinforcement-learning paradigm with reward $-\log(1 - D_\phi(s, G_\xi(s)))$. We use the model-based variant of GAIL (MGAIL) [6] which renders this reward fully differentiable with respect to $\xi$ over a simulation rollout, allowing efficient model training. GAIL has been validated by Kuefler et al. [33] to realistically mimic human-like driving behavior from the NGSim dataset across multiple metrics. These include the similarity of low-level actions (speeds, accelerations, turn-rates, jerks, and time-to-collision), as well as higher-level behaviors (lane change rate, collision rate, hard-brake rate, etc). See Appendix C for a reference to an example video of the learned model driving in a scenario alongside data traces from human drivers.

Our importance sampling and cross-entropy methods use not just a single instance of model parameters $\xi$, but rather a distribution over them to form a generative model of human driving behavior. To model this distribution, we use a (multivariate normal) parametric bootstrap over a trained ensemble of generators $\xi^i$, $i = 1, \ldots, m$. Our models $\xi^i$ are high-dimensional ($\xi \in \mathbb{R}^d$, $d > m$) as they characterize the weights of large neural networks, so we employ the graphical lasso [15] to fit the inverse covariance matrix for our ensemble. This approach to modeling uncertainty in neural-network weights is similar to the bootstrap approach of Osband et al. [38]. Other approaches include using dropout for inference [16] and variational methods [18, 8, 31].

While several open source driving simulators have been proposed [14, 48, 39], our problem formulation requires unique features to allow sampling from a continuous distribution of driving policies for environmental agents. Conditional on each sample of model parameters $\xi$, the simulator constructs a (random) rollout of vehicle behaviors according to $G_\xi$. Unlike other existing simulators, ours is designed to efficiently execute and update these policies as new samples $\xi$ are drawn for each rollout.

## 3.2 System architecture

The second key characteristic of our framework is that it enables black-box testing the AV as a *whole system*. Flaws in complex systems routinely occur at poorly specified interfaces between components, as interactions between processes can induce unexpected behavior. Consequently, solely testing subcomponents of an AV control pipeline separately is insufficient [1]. Moreover, it is increasingly common for manufacturers to utilize software and hardware artifacts for which they do not have any whitebox model [19, 12]. We provide a concise but extensible language-agnostic interface to our benchmark world model so that common AV sensors such as cameras and lidar can provide the necessary inputs to induce vehicle actuation commands.

Our simulator is a distributed, modular framework, which is necessary to support the inclusion of new AV systems and updates to the environment-vehicle policies. A benefit of this design is that simulation rollouts are simple to parallelize. In particular, we allow instantiation of multiple simulations simultaneously, without requiring that each include the entire set of components. For example, a desktop may support only one instance of Unreal Engine but could be capable of simulating

10 physics simulations in parallel; it would be impossible to fully utilize the compute resource with a monolithic executable wrapping all engines together. Our architecture enables instances of the components to be distributed on heterogeneous GPU compute clusters while maintaining the ability to perform meaningful analysis locally on commodity desktops. In Appendix A, we detail our scenario specification, which describes how Algorithm 1 maps onto our distributed architecture.

## 4   Experiments

In this section, we demonstrate our risk-based framework on a multi-agent highway scenario. As the rare-event probability of interest $p_\gamma$ gets smaller, the cross-entropy method learns to sample more rare events compared to naive Monte Carlo sampling; we empirically observe that the cross-entropy method produces 2-20 times as many rare events as its naive counterpart. Our findings hold across different ego-vehicle policies, base distributions $P_0$, and scenarios.

To highlight the modularity of our simulator, we evaluate the rare-event probability $p_\gamma$ on two different ego-vehicle policies. The first is an instantiation of an imitation learning (non-vision) policy which uses lidar as its primary perceptual input. Secondly, we investigate a vision-based controller (vision policy), where the ego-vehicle drives with an end-to-end highway autopilot network [9], taking as input a rendered image from the simulator (and lidar observations) and outputting actuation commands. See Appendix B for a summary of network architectures used.

We consider a scenario consisting of six agents, five of which are considered part of the environment. The environment vehicles' policies follow the distribution learned in Section 3.1. All vehicles are constrained to start within a set of possible initial configurations consisting of pose and velocity, and each vehicle has a goal of reaching the end of the approximately 2 km stretch of road. Fig. 1 shows one such configuration of the scenario, along with rendered images from the simulator. We create scene geometry based on surveyors' records and photogrammetric reconstructions of satellite imagery of the portion of I-80 in Emeryville, California where the traffic data was collected [36].

**Simulation parameters**   We detail our postulated base distribution $P_0$. Letting $m$ denote the number of vehicles, we consider the random tuple $X = (S, T, W, V, \xi)$ as our simulation parameter where the pair $(S, T) \in \mathbb{R}_+^{m \times 2}$ indicates the two-dimensional positioning of each vehicle in their respective lanes (in meters), $W$ the orientation of each vehicle (in degrees), and $V$ the initial velocity of each vehicle (in meters per second). We use $\xi \in \mathbb{R}^{404}$ to denote the weights of the last layer of the neural network trained to imitate human-like driving behavior. Specifically, we set $S \sim 40\text{Beta}(2, 2) + 80$ with respect to the starting point of the road, $T \sim 0.5\text{Beta}(2, 2) - 0.25$ with respect to the lane's center, $W \sim 7.2\text{Beta}(2, 2) - 3.6$ with respect to facing forward, and $V \sim 10\text{Beta}(2, 2) + 10$. We assume $\xi \sim \mathcal{N}(\mu_0, \Sigma_0)$, with the mean and covariance matrices learned via the ensemble approach outlined in Section 3.1. The neural network whose last layer is parameterized by $\xi$ describes the policy of environment vehicles; it takes as input the state of the vehicle and lidar observations of the surrounding environment (see Appendix B for more details). Throughout this section, we define our measure of safety $f : \mathcal{X} \to \mathbb{R}$ as the minimum time-to-collision (TTC) over the simulation rollout. We calculate TTC from the center of mass of the ego vehicle; if the ego-vehicle's body crashes into obstacles, we end the simulation before the TTC can further decrease (see Appendix A for details).

**Cross-entropy method**   Throughout our experiments, we impose constraints on the space of importance samplers (adversarial distributions) for feasibility. Numerical stability considerations predominantly drive our hyperparameter choices. For model parameters $\xi$, we also constrain the search space to ensure that generative models $G_\xi$ maintain reasonably realistic human-like policies (recall Sec. 3.1). For $S, T, W$, and $V$, we let $\{\text{Beta}(\alpha, \beta) : \alpha, \beta \in [1.5, 7]\}$ be the model space over which the cross-entropy method searches, scaled and centered appropriately to match the scale of the respective base distributions. We restrict the search space of distributions over $\xi \in \mathbb{R}^{404}$ by searching over $\{\mathcal{N}(\mu, \Sigma_0) : \|\mu - \mu_0\|_\infty \leq .01\}$, where $(\mu_0, \Sigma_0)$ are the parameters of the base (bootstrap) distribution. For our importance sampling distribution $P_\theta$, we use products of the above marginal distributions. These restrictions on the search space mitigate numerical instabilities in computing likelihood ratios within our optimization routines, which is important for our high-dimensional problems.

We first illustrate the dependence of the cross-entropy method on its hyperparameters. We choose to use a non-vision ego-vehicle policy as a test bed for hyperparameter tuning, since this allows us to take advantage of the fastest simulation speeds for our experiments. We focus on the effects (in

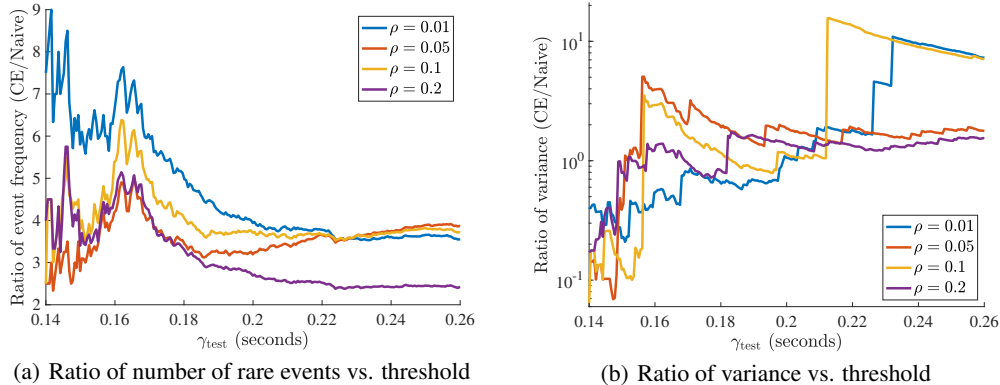

(a) Ratio of number of rare events vs. threshold    (b) Ratio of variance vs. threshold

**Figure 2.** The ratio of $(a)$ number of rare events and $(b)$ variance of estimator for $p_\gamma$ between cross-entropy method and naive MC sampling for the non-vision ego policy. Rarity is inversely proportional to $\gamma$, and, as expected, we see the best performance for our method over naive MC at small $\gamma$.

| Search Algorithm | $\gamma_{\text{test}} = 0.14$ | $\gamma_{\text{test}} = 0.15$ | $\gamma_{\text{test}} = 0.19$ | $\gamma_{\text{test}} = 0.20$ |
|---|---|---|---|---|
| Naive 1300K | $(12.4\pm3.1)$e-6 | $(80.6\pm7.91)$e-6 | $(133\pm3.2)$e-5 | $(186\pm3.79)$e-5 |
| Cross-entropy 100K | $(19.8\pm8.88)$e-6 | $(66.1 \pm 15)$e-6 | $(108\pm 9.51)$e-5 | $(164 \pm 14)$e-5 |
| Naive 100K | $(20\pm14.1)$e-6 | $(100\pm 31.6)$e-6 | $(132\pm11.5)$e-5 | $(185\pm13.6)$e-5 |

**Table 1.** Estimate of rare-event probability $p_\gamma$ (non-vision ego policy) with standard errors. For the cross-entropy method, we show results for the learned importance sampling distribution with $\rho = 0.01$.

Algorithm 1) of varying the most influential hyperparameter, $\rho \in (0, 1]$, which is the quantile level determining the rarity of the observations used to compute the importance sampler $\theta_k$. Intuitively, as $\rho$ approaches 0, the cross-entropy method learns importance samplers $P_\theta$ that up-weight unsafe regions of $\mathcal{X}$ with lower $f(x)$, increasing the frequency of sampling rare events (events with $f(X) \leq \gamma$). In order to avoid overfitting $\theta_k$ as $\rho \to 0$, we need to increase $N_k$ as $\rho$ decreases. Our choice of $N_k$ is borne out of computational constraints as it is the biggest factor that determines the run-time of the cross-entropy method. Consistent with prior works [44, 24], we observe empirically that $\rho \in [0.01, 0.2]$ is a good range for the values of $N_k$ deemed feasible for our computational budget ($N_k = 1000 \sim 5000$). We fix the number of iterations at $K = 100$, number of samples taken per iteration at $N_k = 5000$, step size for updates at $\alpha_k = 0.8$, and $\gamma = 0.14$. As we see below, we consistently observe that the cross-entropy method learns to sample significantly more rare events, despite the high-dimensional nature ($d \approx 500$) of the problem.

To evaluate the learned parameters, we draw $n = 10^5$ samples from the importance sampling distribution to form an estimate of $p_\gamma$. In Figure 2, we vary $\rho$ and report the relative performance of the cross-entropy method compared to naive Monte Carlo sampling. Even though we set $\gamma = 0.14$ in Algorithm 1, we evaluate the performance of all models with respect to multiple threshold levels $\gamma_{\text{test}}$. We note that as $\rho$ approaches 0, the cross-entropy method learns to frequently sample increasingly rare events; the cross-entropy method yields 3-10 times as many dangerous scenarios, and achieves 2-16 times variance reduction depending on the threshold level $\gamma_{\text{test}}$. In Table 1, we contrast the estimates provided by naive Monte Carlo and the importance sampling estimator provided by the cross-entropy method with $\rho = 0.01$; to form a baseline estimate, we run naive Monte Carlo with $1.3 \cdot 10^6$ samples. For a given number of samples, the cross-entropy method with $\rho = 0.01$ provides more precise estimates for the rare-event probability $p_\gamma \approx 10^{-5}$ over naive Monte Carlo.

We now leverage the tuned hyperparameter ($\rho = 0.01$) for our main experiment: evaluating the probability of a dangerous event for the vision-based ego policy. We find that the hyperparameters for the cross-entropy method generalize, allowing us to produce good importance samplers for a very different policy without further tuning. Based on our computational budget (with our current implementation, vision-based simulations run about 15 times slower than simulations with only non-vision policies), we choose $K = 20$ and $N_k = 1000$ for the cross-entropy method to learn a good importance sampling distribution for the vision-based policy (although we also observe similar behavior for $N_k$ as small as 100). In Figure 3, we illustrate again that the cross-entropy method learns to sample dangerous scenarios more frequently (Figure 3a)—up to 18 times that of naive Monte

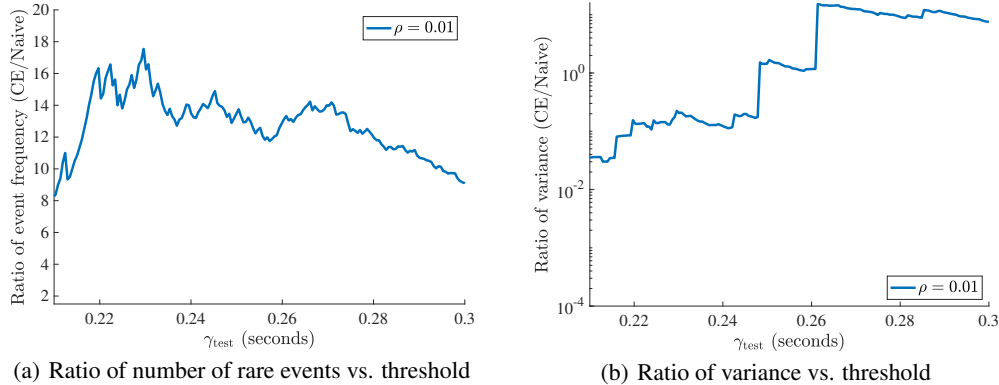

(a) Ratio of number of rare events vs. threshold      (b) Ratio of variance vs. threshold

**Figure 3.** The ratio of (*a*) number of rare events and (*b*) variance of estimator for $p_\gamma$ between cross-entropy method and naive MC sampling for the vision-based ego policy.

| Search Algorithm | $\gamma_{\text{test}} = 0.22$ | $\gamma_{\text{test}} = 0.23$ | $\gamma_{\text{test}} = 0.24$ | $\gamma_{\text{test}} = 0.25$ |
|---|---|---|---|---|
| Cross-entropy 50K | (5.87±1.82)e-5 | (13.0± 2.94)e-5 | (19.0 ± 3.14)e-5 | (4.52 ± 1.35)e-4 |
| Naive 50K | (11.3±4.60)e-5 | (20.6±6.22)e-5 | (43.2±9.00)e-5 | (6.75±1.13)e-4 |

**Table 2.** Estimate of rare-event probability $p_\gamma$ (non-vision ego policy) with standard errors. For the cross-entropy method, we show results for the learned importance sampling distribution with $\rho = 0.01$.

Carlo—and produces importance sampling estimators with lower variance (Figure 3b). As a result, our estimator in Table 2 is better calibrated compared to that computed from naive Monte Carlo.

**Qualitative analysis** We provide a qualitative interpretation for the learned parameters of the importance sampler. For initial velocities, angles, and positioning of vehicles, the importance sampler shifts environmental vehicles to box in the ego-vehicle and increases the speeds of trailing vehicles by 20%, making accidents more frequent. We also observe that the learned distribution for initial conditions have variance 50% smaller than that of the base distribution, implying concentration around adversarial conditions. Perturbing the policy weights $\xi$ for GAIL increases the frequency of risky high-level behaviors (lane-change rate, hard-brake rate, etc.). An interesting consequence of using our definition of TTC from the center of the ego vehicle (cf. Appendix A) as a measure of safety is that dangerous events $f(X) \leq \gamma_{\text{test}}$ (for small $\gamma_{\text{test}}$) include frequent sideswiping behavior, as such accidents result in smaller TTC values than front- or rear-end collisions. See Appendix C for a reference to supplementary videos that exhibit the range of behavior across many levels $\gamma_{\text{test}}$. The modularity of our simulation framework easily allows us to modify the safety objective to an alternative definition of TTC or even include more sophisticated notions of safety, *e.g.* temporal-logic specifications or implementations of responsibility-sensitive safety (RSS) [49, 40].

## 5    Related work and conclusions

Given the complexity of AV software and hardware components, it is unlikely that any single method will serve as an oracle for certification. Many existing tools are complementary to our risk-based framework. In this section, we compare and contrast representative results in testing, verification, and simulation.

AV testing generally consists of three paradigms. The first, largely attributable to regulatory efforts, uses a finite set of basic competencies (*e.g.* the Euro NCAP Test Protocol [46]); while this methodology is successful in designing safety features such as airbags and seat-belts, the non-adaptive nature of static testing is less effective in complex software systems found in AVs. Alternatively, real-world testing—deployment of vehicles with human oversight—exposes the vehicle to a wider variety of unpredictable test conditions. However, as we outlined above, these methods pose a danger to the public and require prohibitive number of driving hours due to the rare nature of accidents [29]. Simulation-based falsification (in our context, simply finding any crash) has also been successfully utilized [51]; this approach does not maintain a link to the likelihood of the occurrence of a particular event, which we believe to be key in acting to prioritize and correct AV behavior.

Formal verification methods [34, 2, 47, 37] have emerged as a candidate to reduce the intractability of empirical validation. A verification procedure considers whether the system can *ever* violate a specification and returns either a proof that there is no such execution or a counterexample. Verification procedures require a white-box description of the system (although it may be abstract), as well as a mathematically precise specification. Due to the impossibility of certifying safety in *all* scenarios, these approaches [49] require further specifications that assign blame in the case of a crash. Such assignment of blame is impossible to completely characterize and relies on subjective notions of fault. Our risk-based framework allows one to circumvent this difficulty by only using a measure of safety that does not assign blame (e.g. TTC) and replacing the specifications that assign blame with a probabilistic notion of how likely the accident is. While this approach requires a learned model of the world $P_0$—a highly nontrivial statistical task in itself—the adaptive importance sampling techniques we employ can still efficiently identify dangerous scenarios even when $P_0$ is not completely accurate. Conceptually, we view verification and our framework as complementary; they form powerful tools that can evaluate safety *before* deploying a fleet for real-world testing.

Even given a consistent and complete notion of blame, verification remains highly intractable from a computational standpoint. Efficient algorithms only exist for restricted classes of systems in the domain of AVs, and they are fundamentally difficult to scale. Specifically, AVs—unlike previous successful applications of verification methods to application domains such as microprocessors [5]—include both continuous and discrete dynamics. This class of dynamics falls within the purview of hybrid systems [35], for which exhaustive verification is largely undecidable [20].

Verifying individual components of the perception pipeline, even as standalone systems, is a nascent, active area of research (see [3, 13, 7] and many others). Current subsystem verification techniques for deep neural networks [28, 30, 50] do not scale to state-of-the-art models and largely investigate the robustness of the network with respect to small perturbations of a single sample. There are two key assumptions in these works; the label of the input is unchanged within the radius of allowable perturbations, and the resulting expansion of the test set covers a meaningful portion of possible inputs to the network. Unfortunately, for realistic cases in AVs it is likely that perturbations to the state of the world which in turn generates an image *should* change the label. Furthermore, the combinatorial nature of scenario configurations casts serious doubt on any claims of coverage.

In our risk-based framework, we replace the complex system specifications required for formal verification methods with a model $P_0$ that we learn via imitation-learning techniques. Generative adversarial imitation learning (GAIL) was first introduced by Ho and Ermon [22] as a way to directly learn policies from data and has since been applied to model human driving behavior by Kuefler et al. [33]. Model-based GAIL (MGAIL) is the specific variant of GAIL that we employ; introduced by Baram et al. [6], MGAIL's generative model is fully differentiable, allowing efficient model training with standard stochastic approximation methods.

The cross-entropy method was introduced by Rubinstein [43] and has attracted interest in many rare-event simulation scenarios [44, 32]. More broadly, it can be thought of as a model-based optimization method [24–26, 53, 27, 56]. With respect to assessing safety of AVs, the cross-entropy method has recently been applied in simple lane-changing and car-following scenarios in two dimensions [54, 55]. Our work significantly extends these works by implementing a photo-realistic simulator that can assess the deep-learning based perception pipeline along with the control framework. We leave the development of rare-event simulation methods that scale better with dimension as a future work.

To summarize, a fundamental tradeoff emerges when comparing the requirements of our risk-based framework to other testing paradigms, such as real-world testing or formal verification. Real-world testing endangers the public but is still in some sense a gold standard. Verified subsystems provide evidence that the AV should drive safely even if the estimated distribution shifts, but verification techniques are limited by computational intractability as well as the need for both white-box models and the completeness of specifications that assign blame (*e.g.* [49]). In turn, our risk-based framework is most useful when the base distribution $P_0$ is accurate, but even when $P_0$ is misspecified, our adaptive importance sampling techniques can still efficiently identify dangerous scenarios, especially those that may be missed by verification methods assigning blame. Our framework offers significant speedups over real-world testing and allows efficient evaluation of black-box AV input/output behavior, providing a powerful tool to aid in the design of safe AVs.

**Acknowledgments**

MOK was partially supported by a National Science Foundation Graduate Research Fellowship. AS was partially supported by a Stanford Graduate Fellowship and a Fannie & John Hertz Foundation Fellowship. HN was partially supported by a Samsung Fellowship and the SAIL-Toyota Center for AI Research. JD was partially supported by the National Science Foundation award NSF-CAREER-1553086.

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
