[Supplementary Material]

# A  Scenario specification

A scenario specification consists of a scenario description and outputs both $p_\gamma$ (1), the accident rate, and a dataset consisting of initial conditions and the minimum time to collision, our continuous objective safety measure. Concretely, a scenario description includes

- a set of possible initial conditions, e.g. a range of velocities and poses for each agent
- a safety measure specification for the ego agent,
- a generative model of environment policies, an ego vehicle model,
- a world geometry model, *e.g.* a textured mesh of the static scene in which the scenario is to take place.

Given the scenario description, the search module creates physics and rendering engine worker instances, and Algorithm 1 then adaptively searches through many perturbations of conditions in the scenario, which we call scenario realizations. A set of scenario realizations may be mapped to multiple physics, rendering, and agent instantiations, evaluated in parallel, and reduced by a sink node which reports a measure of each scenarios performance relative to the specification.

In our implementation the safety measure is minimum time-to-collision (TTC). TTC is defined as the time it would take for two vehicles to intercept one another given that they each maintain their current heading and velocity [52]. The TTC between the ego-vehicle and vehicle $i$ is given by

$$TTC_i(t) = -\frac{r_i(t)}{\dot{r}_i(t)}, \tag{4}$$

where $r_i$ is the distance between the ego vehicle and vehicle $i$, and $\dot{r}_i$ the time derivative of this distance (which is simply computed by projecting the relative velocity of vehicle $i$ onto the vector between the vehicles' poses).

In this paper, vehicles are described as oriented rectangles in the 2D plane. Since we are interested in the time it would take for the ego-vehicle to intersect the polygonal boundary of another vehicle on the road, we utilize a finite set of range and range measurements in order to approximate the TTC metric. For a given configuration of vehicles, we compute $N$ uniformly spaced angles $\theta_1, \ldots, \theta_N$ in the range $[0, 2\pi]$ with respect to the ego vehicle's orientation and cast rays outward from the center of the ego vehicle. For each direction we compute the distance which a ray could travel before intersecting one of the $M$ other vehicles in the environment. These form $N$ range measurements $s_1, \ldots, s_N$. Further, for each ray $s_i$, we determine which vehicle (if any) that ray hit; projecting the relative velocity of this vehicle with respect to ego vehicle gives the range-rate measurement $\dot{s}_i$. Finally, we approximate the minimum TTC for a given simulation rollout $X$ of length $T$ discrete time steps by:

$$f(X) := \min_{t=0,\ldots,T} \left( \min_{i=1,\ldots,N} \frac{-s_i(t)}{\dot{s}_i(t)} \right)$$

Note that this measure can approximate the true TTC arbitrarily well via choice of $N$ and the discretization of time used by the simulator. Furthermore, note that our definition of TTC is with respect to the *center* of the ego vehicle touching the *boundary* of another vehicle. Crashing, on the other hand, is defined in our simulation as the intersection of boundaries of two vehicles. Thus, TTC values we evaluate in our simulation are nonzero even during crashes, since the center of the ego vehicle has not yet collided with the boundary of another vehicle.

# B  Network architectures

The MGAIL generator model we use takes the same inputs as that of Kuefler et al. [33]—the dynamical states of the vehicle as well as virtual lidar beam reflections. Specifically, we take as inputs: geometric parameters (vehicle length/width), dynamical states (vehicle speed, lateral and angular offsets with respect to the center and heading of the lane, distance to left and right lane boundaries, and local lane curvature), three indicators for collision, road departure, and traveling in reverse, and lidar sensor observations (ranges and range-rates of 20 lidar beams) as depicted in Figure 4. The generator has two hidden layers of 200 and 100 neurons. The output consists of the mean and variance of normal distributions for throttle and steering commands; we then sample from

**Figure 4:** Depiction of lidar sensor input used for GAIL models.

these distributions to draw a given vehicle's action. The discriminator shares the same size for hidden layers. The forward model used to allow fully-differentiable training first encodes both the state and action through a 150 neuron layer and also adds a GRU layer to the state encoding. A Hadamard product of the results creates a joint embedding which is put through three hidden layers each of 150 neurons. The output is a prediction of the next state.

The end-to-end highway autopilot model is a direct implementation of Bojarski et al. [9] via the code found at the link `https://github.com/sullychen/autopilot-tensorflow`. In our implementation of the vision-based policy, this highway autopilot model uses rendered images to produce steering commands. Lidar inputs are used to generate throttle commands using the same network as the non-vision policy.

## C  Supplementary videos

We have provided some videos to augment the analysis in our paper (available in the NeurIPS supplement and at `http://amansinha.org/docs/OKellySiNaDuTe18_videos.zip`):

- gail.mp4 provides an example of a trained GAIL model driving alongside data traces from real human drivers [36].

- Example videos from rollouts. The filenames start with "mttc =" to indicate the minimum TTC that resulted between the ego and any other vehicle during the rollout. Note that even crashes have nonzero values of TTC due to the definition we used for TTC from the center of the ego vehicle (cf. Appendix A). The videos are all played back at $2.5\times$ real-time speed. The videos included in the supplement are:
  - Crashes:
    * mttc $= 0.23 -$ crash.mp4
    * mttc $= 0.30$.mp4
    * mttc $= 0.42$.mp4
    * mttc $= 0.56$.mp4
  - Non-crashes:
    * mttc $= 0.23 -$ nocrash.mp4
    * mttc $= 0.79$.mp4
    * mttc $= 1.43$.mp4
    * mttc $= 2.01$.mp4
    * mttc $= 3.05$.mp4
    * mttc $= 6.00$.mp4
    * mttc $= 6.01$.mp4
    * mttc $= 10.11$.mp4

  These videos contain overhead, RGB, segmented, and depth views. We also include higher-resolution RGB videos with the same base names as above but the extension "_hires.mp4".