[Reviews · NeurIPS 2018]

Reviewer 1



Summary: In this work, the authors study the problem of testing autonomous vehicles. Real-world testing is putting public into risk and the rare nature of accidents require billions of miles to evaluate. The authors implement a simulation framework to test an entire autonomous driving system. Their goal is to evaluate the probability of an accident under a base distribution governing standard traffic behavior. The demonstration of this framework shows that it is possible to accelerate system evaluation by 10-50P times that of real world testing and 1.5-5 times that of naive Monte Carlo sampling methods. Strong Points: 1) Interesting problem and solution. The authors try to solve the problem of AV testing by taking into consideration the two following challenges: the rare nature of serious accidents, how to verify the correctness of an AV? 2) The paper introduced interesting ideas on how to improve simulations for AV testing. 3) Claims are well proved, discussed and explained. 4) Extended literature review on verification methods, perception pipeline, cross-entropy method, and generative adversarial imitation learning. Weak Points: 1) The authors should include which methods are used now for testing AVs in a related work section. 2) Evaluation of the proposed framework is not very clear. 3) There is no explanation on how to pick values for the parameters in the Experiments section. Minor Comments: 1) Line 117: The word “we” should be deleted. 2) Line 118: The word “use” is missing. 3) Line 485: The word "followed" is misspelled. *** After reading author's feedback *** I have read authors' responses, I like the paper and I am happy to see that our minor comments were taken under consideration.

Reviewer 2



Very strong paper which is reporting on a large scale AV project. As such it describes many parts of a very complex system, relying on a lot of other work and systems, but contributing by showing how it all fits together to do a new thing. The new thing is the use of importance sampling over parameters for models of human drivers in a driving simulator, in order to better estimate collision probabilities for AV controllers. This relies on a huge stack of prior work comprising: (1) a complex physical driving simulator (which brilliantly, is open sourced, and presentation of such a facility to the community would make a great publication just by itself); and (2) a parametric model of human driver behaviour based on a GAN-style imitation learner, which maps an input vector describing the scene to a prediction of other drivers actions. Paper is clearly written and structured, however for slightly dumber readers like this one I would suggest trying to add a bit more detail on exactly what are the inputs and outputs of this human driver predictor. From my reading: I /think/ they are a mixture of geometric numbers describing distances to other vehicles and similar, plus a load of raw lidar (and other?) sensor data for the deep learner to do its thing with. But predicting humans is really, really, hard, and so I'd like to be more certain about exactly what these are in order to see what real-world inputs of interest might be missing. I'd also like a clearer presentation of how accuracte the predictions are for the human drivers seen in a test set of the input data; there are arguments that human prediction of other drivers is based on all manner of psychology relevant input features so I want to know exactly what is the prediction accuracy here (maybe compared to how well human video watchers can do too if possible?). Given the high quality of the paper its possible this is already mentioned but if so please point me to it and/or make clearer in the text; otherwise please add. Impressively again, these models are also promised as fully open source (though not yet for anonymity reasons) which sounds EXCELLENT. Given all of the above stack, we then come to the actual research contribution of the paper, which is the use of importance sampling to estimate the craqsh probability for controllers. Importance sampling is of course quite standard but doing it on this scale and on such complex models is interesting to see, and appears to be a useful and important application to a real problem. So is a great paper for three main reasons (1) as an update on a big ongoing project for this community; (2) as a provider of some big and useful open source tools for the community; and (3) for the interesting and useful use of importance sampling over a novel simulation and model. Good to see my suggestion for evaluation addresses in the author response, and to see the confirmation that everything will be open-sourced.

Reviewer 3



This paper introduces a simulation framework -- called a pseudoreality -- that can test an entire autonomous driving system, including deep-learning perception + control algorithms, underlying dynamics models and a photo-realistic rendering engine. They demonstrate full-scale testing using a risk-based framework, evaluating the probability of an accident under some base distribution that stipulates traffic behavior. Adaptive importance sampling methods are used to evaluate risk. The authors demonstrate that system testing can be done by 10-50 times the number of processors used in real-world testing and 1.5-5 times faster than that of naive MC sampling. The problem domain examined in this paper is one that is extremely relevant, given the introductions of autonomous vehicles onto roads, both in city and highway driving. Due to the rare nature of accidents and uncertainty in real-world environments, testing is computationally prohibitive and so some verification in simulated environments is needed to ensure safety and performance guarantees. A probabilistic "risk-based" framework that simulates rare events only makes sense. The authors try to estimate the probability of a rare event via simulations and generate scenarios that exhibit risky behavior (defined by a threshold parameter). The key premise of the proposed simulator requires that a model of the world be estimated. This is done by using public traffic data collected by the US DOT (Department of Transportation). Imitation learning is used to learn a generative model for the behavior/policy of environment vehicles, resulting in an ensemble of models that characterize the distribution of human-like driving policies. Specifically, an ensemble of generative adversarial imitation learning models are used to learn the base distribution. To generate risky scenarios and to estimate the probability of a risky event, the authors use adaptive importance sampling (instead of naive MC sampling) to speed up evaluations. They propose a cross-entropy algorithm that iteratively approximates the optimal importance sampling distribution. Importance sampling + cross-entropy methods choose a distribution over the model parameters in the generative model of human driving behavior. The simulator then constructs a random rollout of behaviors according to the learned policy of a human driver given environmental inputs. The simulator, then, is the primary contribution of this paper, which is supported by the risk-based framework and cross-entropy search techniques. It can take inputs such as an rendered image and output actuated commands based on an end-to-end highway autopilot network. Testing of the simulator is done using a multi-agent highway scenario where public traffic data is available. The authors show that their simulator accelerates the assessment of rare-event probabilities with respect to real-world testing. The ideas presented in this paper are interesting in that they attempt to address the problem of how to test and ensure safety and performance of autonomous driving vehicles. Combining adaptive importance sampling + cross-entropy methods in a simulator that attempts to estimate and generate risky events seems like a reasonable approach. From the paper's exposition, it appears that this is a novel idea and, if so, would be certainly of interest to the machine learning community. I have read the authors' response and thank them for their comments.